# Tumor Microenvironment as a Therapeutic Target in Melanoma Treatment

**DOI:** 10.3390/cancers15123147

**Published:** 2023-06-11

**Authors:** Naji Kharouf, Thomas W. Flanagan, Sofie-Yasmin Hassan, Hosam Shalaby, Marla Khabaz, Sarah-Lilly Hassan, Mosaad Megahed, Youssef Haikel, Simeon Santourlidis, Mohamed Hassan

**Affiliations:** 1Biomaterials and Bioengineering, Institut National de la Santé et de la Recherche Médicale, Université de Strasbourg, Unité Mixte de Recherche 1121, 67000 Strasbourg, France; dentistenajikharouf@gmail.com (N.K.); youssef.haikel@unistra.fr (Y.H.); 2Department of Endodontics and Conservative Dentistry, Faculty of Dental Medicine, University of Strasbourg, 67000 Strasbourg, France; 3Department of Pharmacology and Experimental Therapeutics, LSU Health Sciences Center, New Orleans, LA 70112, USA; tflan1@lsuhsc.edu; 4Department of Chemistry, Faculty of Science, Heinrich-Heine University Duesseldorf, 40225 Dusseldorf, Germany; sofie00@gmx.de (S.-Y.H.); slh03122001@gmail.com (S.-L.H.); 5Department of Urology, School of Medicine, Tulane University, New Orleans, LA 70112, USA; 6Department of Production, Beta Factory for Veterinary Pharmaceutical Industries, Damascus 0100, Syria; marla.khabaz2@gmail.com; 7Clinic of Dermatology, University Hospital of Aachen, 52074 Aachen, Germany; mmegahed@ukaachen.de; 8Pôle de Médecine et Chirurgie Bucco-Dentaire, Hôpital Civil, Hôpitaux Universitaire de Strasbourg, 67000 Strasbourg, France; 9Epigenetics Core Laboratory, Institute of Transplantation Diagnostics and Cell Therapeutics, Medical Faculty, Heinrich-Heine University Duesseldorf, 40225 Duesseldorf, Germany; simeon.santourlidis@med.uni-duesseldorf.de; 10Research Laboratory of Surgery-Oncology, Department of Surgery, School of Medicine, Tulane University, New Orleans, LA 70112, USA

**Keywords:** tumor microenvironment, stromal cells, melanoma resistance, targeted therapy, chronic inflammation

## Abstract

**Simple Summary:**

As a solid tumor, melanoma is not only a tumor mass of monolithic tumor cells, but it also contains supporting stroma, extracellular matrix (ECM), and soluble molecules forming the widely recognized tumor microenvironment. The main components of the tumor microenvironment include stromal cells (endothelial cells, fibroblasts, mesenchymal stem cells, and immune cells), ECM, and soluble molecules (chemokines, cytokines, growth factors, and extracellular vesicles). The tumor microenvironment has been suggested to play a central role in tumor progression and treatment resistance. Accumulated evidence indicates that tumor maintenance, progression, and treatment resistance are determined by components of the microenvironment. Thus, targeting the components of the tumor microenvironment may have a therapeutic impact on melanoma treatment. The main topic of this paper deals with the main components of the tumor microenvironment and their impact as therapeutic targets in melanoma treatment.

**Abstract:**

The role of the tumor microenvironment in tumor growth and therapy has recently attracted more attention in research and drug development. The ability of the microenvironment to trigger tumor maintenance, progression, and resistance is the main cause for treatment failure and tumor relapse. Accumulated evidence indicates that the maintenance and progression of tumor cells is determined by components of the microenvironment, which include stromal cells (endothelial cells, fibroblasts, mesenchymal stem cells, and immune cells), extracellular matrix (ECM), and soluble molecules (chemokines, cytokines, growth factors, and extracellular vesicles). As a solid tumor, melanoma is not only a tumor mass of monolithic tumor cells, but it also contains supporting stroma, ECM, and soluble molecules. Melanoma cells are continuously in interaction with the components of the microenvironment. In the present review, we focus on the role of the tumor microenvironment components in the modulation of tumor progression and treatment resistance as well as the impact of the tumor microenvironment as a therapeutic target in melanoma.

## 1. Introduction

Melanoma is one of the most common skin cancers, and it is notorious for its heterogeneity and propensity to metastasize to distant organs [1,2]. Although the treatment options of melanoma have improved in recent years, patients with advanced malignant melanoma still have poor prognosis, as measured by progression-free and overall survival [3].

Molecularly targeted therapies are characterized by their specificity to interfere with key molecules of aberrant signaling pathways, particularly those of tumor growth and survival. Over 60% of primary cutaneous melanomas and over 50% of metastatic melanomas harbor the activating murine sarcoma viral oncogene homolog B (BRAF) mutation [4,5]. To that end, the continuous activation of mitogen-activated protein kinase (MAPK)/extracellular signal-regulated kinase (ERK) signaling by the BRAF^V600E^ mutation is common and independent from extracellular stimulation [6,7]. Melanoma is a tumor mass that contains supporting stroma (fibroblasts, endothelial cells, and immune cells), the extracellular matrix (ECM), and soluble molecules (chemokines, cytokines, growth factors, and extracellular vesicles), rather than a mass of monolithic tumor cells, as shown in Figure 1. As a result, melanoma cells are continuously in active interaction with the components of the microenvironment. The crosstalk within the tumor microenvironment is the main driver for the generation of the malignant phenotypes in the form of genetic divergent subpopulations with intra- and intertumoral heterogeneity. Here, we will give insight about the role of the tumor microenvironment on melanoma progression and treatment resistance and the impact of the tumor microenvironment as a therapeutic target for melanoma treatment.

## 2. Mechanisms of Melanoma Resistance

Although successful targeting of melanomas with the BRAF^V600E^ mutation improves overall survival, the long-term efficacy of available therapeutics, including BRAF inhibitors (e.g., vemurafenib and dabrafenib alone or in combination with mitogen-activated protein kinase (MEK) inhibitors (e.g., Trametinib and Selumetinib), are unable to cause complete tumor regression [8,9]. In addition to their limitation in tumor regression, current treatment regimens are mostly associated with the occurrence of subsequent mutations within genes encoding key molecules of aberrant signaling pathways.

Melanoma cells with the active mutation BRAF^V600E^ are infamous for their ability to frequently enforce several aberrant molecular and cellular mechanisms conferring resistance to targeted therapeutics. These molecular and cellular mechanisms are mediated by higher mutation rates, alteration of membrane drug transporter mechanisms, enhancement of DNA repair mechanisms, dysregulation of cell death machinery [10,11], and promotion of autophagy [12].

Besides its ability to evade drug toxicity, melanoma can also develop adaptive tumor cell plasticity via tumor-microenvironment-dependent modifications [13,14,15], in response to treatment with BRAF/MEK inhibitors [10,11]. In addition, BRAF/MEK inhibitors-induced reactivation of MAP kinase signaling by the enhancement of BRAF^V600E^ amplification or alternative splicing, RAS-mutation can also contribute to the development of tumor plasticity [10,16]. The role of tumor plasticity in the development of tumor resistance to anticancer agents has been reported in several studies [17,18,19].

Tumor cell plasticity is driven by selective epigenetic and molecular changes that allow transitions within a wide range of cellular phenotypes [20,21,22,23,24]. These reported changes have been shown to drive tumor cell diversity and promote reversible phenotypic spectrums as well. The processes of melanoma plasticity are mediated via microenvironmental components, stochastic genetic and epigenetic alterations, treatment-imposed selective-pressure-dependent mechanisms mediating tumor progression, and treatment resistance [23,24]. To that end, dedifferentiated melanoma cells are infamous for their aggressive characteristics, which lead to metastasis and resistance to anticancer agents.

The phenotypic alterations of melanoma cells range from a differentiated and proliferative cell state to dedifferentiated mesenchymal-like phenotypes with intermediate states [25,26]. Cutaneous malignant melanomas are heterogeneous in nature and comprise several genetically divergent subpopulations with distinct transcriptomic signatures that determine their behaviors [27,28]. Accordingly, melanomas carry different genetic alterations that have different clinical features and disease outcomes. Microphthalmia transcription factor (MITF) is a member of the most important transcriptomic family of melanoma. MITF has been shown to be the master regulator of not only normal melanocytes but also malignant melanomas [29,30]. Based on the expression levels of MITF, two melanoma cell populations have been identified. One of these populations is a fast-replicating population that exhibits high MITF expression associated with low invasive potential; the other population exhibits high MITF expression associated with a slow proliferation rate and high invasive potential [29,31].

The most dynamic trend of MITF in melanoma is the transition to a mesenchymal phenotype and to enhance autophagy in response to a variety of tumor microenvironmental stresses. These microenvironmental stresses include nutrient and oxygen deprivation, inflammation, immune defense, or therapies. Tumor microenvironmental stresses can promote melanoma cell survival and the generation of adaptive phenotypes with ability to evade drug toxicity [32,33,34].

Autophagy drives lysosome-dependent degradation of cytoplasmic components in response to starvation [35,36]. Autophagy is characterized by the ability to influence diverse aspects of homeostasis with inhibition of malignant transformation [12,36]. Apart from its inhibitory role in tumor initiation, autophagy can also mediate survival mechanisms based on its ability to stimulate tumor growth and to confer drug resistance [12,36].

Although the molecular mechanisms of autophagy are orchestrated by *Atg* gene products in the cytoplasm [12,36,37], the regulation of the autophagy process is mediated by the MITF family that encodes four distinct genes: MITF, TFEB, TFE3, and TFEC [38]. Three of the MITF family genes, MITF, TFEB, and TFE3, have been identified as regulators of lysosomal function and metabolism [39,40]. Lysosomal and autophagy genes, particularly those possessing one or more 10 base pair motifs (GTCACGTGAC), are targets for the MITF family as transcription factors [41]. As a result, the expression and regulation of their cognate genes are continuously under the control of MITF family members.

While differentiated melanoma cells exhibit high levels of MITF and SOX10, dedifferentiated melanoma cells show low expression of MITF and high expression of mesenchymal markers [42,43]. Dedifferentiated melanoma cells also exhibit low expression of proliferative and invasive factors [25,44], ECM [45], and resistance markers [44,46]. Accordingly, the most common marker of differentiated melanoma cells is the receptor tyrosine kinase (RTK) AXL [47,48].

Dedifferentiation is a hallmark of cancer progression and is responsible for the development of cross resistance to both targeted and immune therapies in melanoma [49]. The differentiation plasticity of melanoma can be attributed to the embryonic history of melanocytes [50]. Melanocytes are derived from the neural crest, a transient, migratory, and multi-potent population of cells that can differentiate into diverse cell types [51].

Dedifferentiated melanoma cells are characterized by their low expression of proliferative and invasive factors [25,44], ECM [45], and resistance markers [44,46]. The most common marker of differentiated melanoma cells is the receptor tyrosine kinase (RTK) AXL [47,48].

In melanoma cells, the development of acquired resistance is attributed to the downregulation of MITF [52]. In addition to its role in the development of tumor resistance, MITF acts as a master regulator of melanocyte differentiation through the upregulation of receptor tyrosine kinases (RTKs) including the AXL, EGFR, and PDGFRβ [53,54].

The AXL protein is characterized by an extracellular structure consisting of two fibronectin type 3-like repeats and two immunoglobulin-like repeats along with its intracellular tyrosine kinase domain. All members of the TYRO3, AXL, and MER (TAM) tyrosine kinase receptor family are involved in the regulation of cell proliferation, epithelial–mesenchymal transition (EMT), migration, and regulation of immune responses [55]. Additionally, AXL plays an important role in the regulation of downstream signaling via PI3K/AKT, MAPK/ERK, and STAT3 pathways [12,56,57]. Figure 2 outlines the functional role of AXL in the regulation of different cellular processes.

The elevated expression of AXL has been shown in many cancer types including melanoma, lung, breast, and pancreatic cancers [58,59]. Thus, as a tyrosine kinase receptor, AXL functionally plays an essential role in different oncogenic processes in addition to its suppressive activity, which leads to the destruction of cell death machinery in melanoma [59,60].

The role of AXL in melanoma migration and invasion has been demonstrated. The stratification of melanomas into distinct subsets by a gene-signature-dependent study revealed that AXL expression is correlated positively or negatively with identified gene signatures [22,61]. The set of genes that were positively associated with AXL expression were genes associated with extracellular matrix interactions and remodeling, indicating a possible role for AXL in melanoma invasion [62]. The reduction in melanoma invasion and migration following inhibition of AXL by its specific siRNAs or pharmacological inhibition confirms a significant role for AXL in the modulation of tumor progression and migration [63,64]. In addition to its role in melanoma invasion and migration, AXL has been shown to promote both intrinsic and acquired resistance to chemotherapeutic, immunotherapeutic, and molecularly targeted therapies both in solid and hematologic malignancies [54,65]). In malignant melanoma, the high level of AXL is mostly associated with the mesenchymal cell state, leading to enhanced resistance to targeted therapy, such as MAPK inhibitors in the case of malignant melanoma [66,67] and endothelial growth factor receptor (EGFR) inhibitors in case of lung cancer. Elevated expression of AXL also reduces tumor sensitivity to both chemotherapy and poly ADP ribose polymerase (PARP) inhibition [59,68,69], and confers resistance to cisplatin in melanoma cells [60] and in ovarian cancer cells [70].

BRAF inhibitors-induced activation of CAFs has been shown to trigger both tumor progression and treatment resistance in melanoma cells [71]. Apart from the limited therapeutic success of BRAF/MEK inhibitors in melanoma patients, most patients exhibited treatment failure along with the development of acquired resistance [72]. The observed treatment failure in melanoma patients was found to result from BRAF/MEK-inhibitor-induced activation of the SEMA6A/RhoA/YAP axis as consequence of the crosstalk between tumor and stromal cells [72].

BRAF-inhibitor-induced TGF-β release in melanoma cells has been demonstrated to promote the activation of CAFs [73]. Activated CAFs can trigger melanoma progression and resistance to BRAF/MEK inhibitors via the release of soluble molecules including neuregulin 1 and hepatocyte growth factor (HGF) [73,74] as well as fibronectin [75,76].

Fibroblast-derived neuregulin 1 has been shown to trigger activation of ErbB3-receptor-dependent signaling pathways to promote tumor progression and treatment resistance in melanoma patients [74]. Likewise, fibroblast-derived HGF has been shown to enhance epithelial–mesenchymal transition (EMT) and to confer resistance to anticancer agents [75,77]. The contribution of HGF to the promotion of EMT is mediated by the induction of EMT-associated transcription factors, such as Snail1 [78] and Zeb1 [78,79]. The anti-apoptotic activity of Snail1 has been shown to be involved in the development of resistance to immune and targeted therapies [80,81]. CAF-derived fibronectin interacts with and activates cell surface integrin receptors that, in turn, serve to recruit a series of cellular proteins leading to the enhancement and promotion of many cellular functions leading to tumor migration and invasiveness [82,83,84]. The induction of tumor invasiveness by fibronectin is mediated by FAK-induced regulation of MMP-9 via ERK and PI3K pathway-dependent activation. [85]. Therefore, CAF-released fibronectin is expected to play an important role in the promotion of migration and invasiveness of many tumor types, including melanomas [12,82].

Figure 3 demonstrates the mechanisms whereby BRAF inhibitors induce TGF-β release in melanoma cells and the biological consequences of TGF-β on the conversion of fibroblasts into CAFs and the promotion of CAFs to release their specific factors and mediators to trigger the suppression of the adaptive immune system, ECM remodeling, melanoma progression, and treatment resistance.

## 3. Tumor Microenvironment

In contrast to normal tissues, tumors are characterized by their unique microenvironments that determine their fate and behavior during their evolution [86,87]. The unique characteristics of tumor microenvironments are attributed to rapid tumor proliferation and metabolism compared to normal tissues [86]. Concordantly, the tumor microenvironment is more acidic and exhibits high levels of reactive oxygen species (ROS) and glutathione (GSH), higher hypoxic status, overexpressed enzymes, and high levels of ATP. Accordingly, the tumor microenvironment has emerged as an important therapeutic target during tumor treatment.

As mentioned above, solid tumors are tumor masses containing a wide range of non-cancer stromal cells rather than a mass of monolithic tumor cells. The non-stromal cell tumor microenvironment contains a wide range of stromal cells, which can be transformed into cancer-associated stromal cells by tumor-derived growth factors, including platelet-derived growth factor (PDGF) and TGF-β [88,89]. Thus, upon their activation, cancer-associated stromal cells can play an essential role in the modulation of tumor progression and treatment resistance through their released secretory factors.

The most important cell type among tumor-associated stromal cells is the cancer-associated fibroblasts (CAFs). CAFs are characterized by their released secretory molecules, which include cytokines, chemokines, and growth factors, which can promote the deposition and remodeling of the ECM [90,91]. CAF-derived cytokines, chemokines, and growth factors are essential in driving cellular processes that lead to tumor growth, angiogenesis, inflammation, and drug resistance [92,93]. Thus, increase in CAF number in the tumor stroma is mostly associated with poor prognosis and increased risk of metastasis [94,95].

Under normal physiological conditions, the maintenance of the structure and homeostasis of the skin is highly controlled by the crosstalk between normal melanocytes and the surrounding microenvironment, including ECM, keratinocytes, fibroblasts, endothelial cells, and immune cells [96]. The intercellular communications between melanocytes and microenvironment components are mediated mainly through paracrine interactions and/or cell–cell contact via cell adhesion molecules [97,98].

Recent evidence indicates that the most important stromal cell type in connective tissue is the fibroblast [99]. Fibroblasts play an important role in the regulation of many physiological and pathological processes that are involved in the regulation of ECM turnover and homeostasis [100,101], epidermal regeneration [102,103], wound healing [103], and tumor progression and treatment resistance [71,90]. The transformation of fibroblasts into CAFs results from the crosstalk of normal fibroblasts with tumor cells via tumor-cell-released TGF-β which leads to the differentiation of fibroblasts into CAFs as shown in Figure 3. In contrast to normal fibroblasts, CAFs are the main player in the microenvironment of solid tumors, based on their released factors that can drive tumor progression, metastasis, and treatment resistance [104,105].

Activated CAFs secrete more cytokines and chemokines than their resting counterparts and thereby trigger tumor progression and treatment resistance. The most common CAF-released factors are TGF-β, PDGF, FGF, HGF, vascular endothelial growth factor (VEGF), tumor necrosis factor α (TNFα), interferon-γ (IFNγ), CXCL12, IL-6, connective tissue growth factor (CTGFβ), EGF, growth arrest-specific protein 6 (GAS6), galectin-1, secreted frizzled-related protein 1 (SFRP1), sonic hedgehog protein (SHH), bone morphogenetic protein (BMP), fibroblast-specific protein-1 (FSP-1/S100A4), fibroblast-activating protein (FAP), platelet-derived growth factor receptor-alpha/beta (PDGFR α/β), tenascin C, desmin, collagen 11-α1 (COL11A1), vimentin, periostin, and fibronectin [106,107,108]. All these chemokines and growth factors are involved in the modulation of tumor progression, metastasis, resistance, and recurrence.

In addition to its role in the regulation of fibrosis, TGF-β is the major CAF-released factor [109,110]. TGF-β is mainly involved in modulation of the crosstalk between CAFs and cancer cells [111,112]. Therefore, inhibition of TGF-β signaling is mostly associated with tumor growth and metastasis [113].

Although chemotherapy-based treatment can target rapidly proliferating cells, the elimination of CAFs by chemotherapy is a rare occurrence [114,115]. The contribution of CAFs in the development of tumor resistance is common and occurs mostly after treatment has been initiated [116,117]. Many in vitro experiments indicate that anticancer-agent-induced DNA damage is correlated with an increase in cancer cell invasion and survival [118]. DNA-damage-associated tumor invasion and survival seems to be the consequence of stromal-derived paracrine signaling via cytokines and exosome-dependent mechanisms [107].

The interaction between malignant melanoma cells and dermal fibroblasts is responsible for the generation of CAFs that, in turn, mediates melanoma progression, metastasis, and treatment resistance [119,120].

The transformation of resting fibroblasts into CAF-like phenotypes by anticancer agents has been reported [121,122]. The contribution of CAF-like phenotypes in the promotion of stemness properties in tumor cells derived from either breast [123] or colorectal cancers [124,125] has also been reported. CAFs can induce stemness properties of cancer cells via the activation of hypoxia-inducible factor (HIF-1α) and sonic hedgehog-GLI signaling [126,127]. Additionally, CAF-mediated TGF-β signaling has been suggested to synergize with HIF-1α signaling to enhance the expression of GLI2 that subsequently triggers stemness properties in target tumor cells [127,128].

Like many chemotherapeutics, radiation therapy triggers tumor cell death via DNA-damage-dependent mechanisms. The exposure of tumor mass to radiation therapy does not trigger DNA damage in tumor cells nor in stromal cells within the tumor microenvironment. Therefore, irradiated fibroblasts can overcome apoptotic signals and undergo cellular senescence to exhibit a highly activated CAF phenotype [129]. Exposure of activated CAFs to radiation has been shown to trigger CXCL12 overexpression, leading to epithelial-to-mesenchymal transition (EMT) and invasion of tumor cells in vitro and in vivo [130].

Senescence is cell cycle arrest that can be activated by oncogenic signaling. Therefore, senescence can limit tumor progression and can alter the outcome of anticancer therapies. Senescent cells are characterized by stable cell cycle arrest associated with the secretion of several factors termed senescence-associated secretory phenotype (SASP) [131]. Like in other solid tumors, cellular senescence is common in melanoma and has been observed in melanoma patients, particularly during and/or after the treatment course has been initiated [132,133]. Senescence is a mechanism through which many tumor cells overcome the action of anti-cancer agents to survive, grow, and metastasize [134,135]. Cellular senescence is an autonomous tumor suppressor mechanism associated mainly with the stabilization of cell cycle arrest [136]. Senescent cells with the SASP phenotype secrete high levels of inflammatory cytokines, immune modulators, growth factors, and proteases [69,137,138]. Accordingly, SASP triggers significant changes in tumor cells and their microenvironment, enabling the tumor to evade drug toxicity and ultimately grow and relocate to distant organs [139]. Many reports have shown that SASP-dependent mechanisms allow senescent cells to drive a range of different cellular processes [140,141] The formation of SASP is dynamic and spatially regulated [142,143]. Changing of the ingredients of SASP composition can therefore determine the beneficial and detrimental aspects of the senescence program, tipping the balance to either an immunosuppressive/pro-fibrotic environment or pro-inflammatory/fibrolytic state [144]. SASP has the potential to trigger tumor growth through the changing of the tumor microenvironment composition and can influence treatment outcomes. In vivo elimination of senescent cells via senolysis significantly impacts the treatment of aging [145,146]. In contrast to the treatment of aging, the activation of senescence has attracted more attention of researchers and the pharmaceutical industry. Senolytic therapy represents a group of mechanistically diverse drugs that can eliminate the establishment of senescence, which is expected to inhibit tumor development and progression. Senescence occurrence can be caused by either intensive DNA damage, telomere shortening, or oncogene activation [147]. Oncogene-induced senescence is one of the common features of melanocytic nevi that is essential to prevent oncogenesis and malignant transformation of benign lesions [148,149]. Melanocytic nevi are precursors for the development of malignant melanomas [150]. Thus, the stabilization of oncogene-induced senescence or the elimination of senescent melanocytes represents a promising approach to the prevention of tumorigenesis.

Microenvironment targeting of checkpoint proteins by programmed death-1 (PD-1) and cytotoxic T lymphocytes-associated antigen-4 (CTLA-4) inhibitors brought more attention to the crosstalk between the immune cell and tumor microenvironment; however, the regulation of immune cell–tumor crosstalk, so far, is not well described [151,152].

As one of the abundant components of the tumor microenvironment, CAFs have been reported to mediate a tumor immune landscape by the secretion of various soluble molecules including cytokines, growth factors, chemokines, exosomes, and other effector molecules [90,153]. Thus, CAF-released factors are necessary to promote an immunosuppressive tumor microenvironment enabling cancer cells to evade immune surveillance [154,155], a mechanism whereby CAFs limit the efficacy of the immune therapy.

The mechanisms by which CAF-released factors trigger tumor angiogenesis, ECM remodeling, suppression of adaptive immunity, melanoma progression, metastasis, treatment resistance, and stemness properties are outlined in Figure 4.

## 4. Tumor Microenvironment as Therapeutic Target in Melanoma Treatment

Apart from its adverse effects, chemotherapeutic agents remain the best option for cancer therapy to date. Depending on the tumor stage and patient tolerability, chemotherapy can be given alone or in combination with surgery or radiotherapy [156,157]. The discovery of active mutations, which are involved in tumor initiation and development, such as epidermal growth factor receptor (EGFR), p53, c-Kit and BRAF [158,159], compelled researchers to extensively study the reliability of such mutations as selective therapeutic targets [160,161,162]. Although the successful targeting of these mutations improves overall survival of melanoma patients, acquired tumor resistance develops and increases continuously [163,164]. Consequently, tumor relapse and low life quality of patients are common.

During tumor development, cancer cells and the components of the tumor microenvironment are continually adapting to the environmental conditions to promote tumor growth, progression, and treatment resistance [165].

Tumor microenvironment components play a significant role in cancer progression, maintenance, and resistance to anticancer agents [128,166]. The crosstalk between tumor cells and their microenvironment is essential for acquiring and maintaining tumor cell characteristics, such as sustaining proliferative signaling, resisting cell death, inducing angiogenesis, activating invasion, metastasis, triggering tumor-promoting inflammation, and avoiding immune destruction [167,168,169]. This dependence on the tumor microenvironment offers an opportunity for the development of therapeutic approaches by targeting the components of the tumor microenvironment or their dependent signaling pathways. Based on the increased understanding of the crucial roles of the tumor microenvironment on tumor development and therapeutic resistance, many efforts have been made to target tumor microenvironment components for therapeutic benefit in cancer patients [170]. Importantly, targeting the components of the tumor microenvironment has a significant therapeutic advantage over the direct targeting of cancer cells, as cancer cells are infamous for their genomic instability that is a main cause for the development of drug resistance [132,171]. In contrast, the non-tumor cells of the tumor microenvironment are genetically more stable in nature and more susceptible [172].

### 4.1. Cancer-Associated Fibroblasts as Therapeutic Target

Over the recent decade, accumulating evidence revealed that CAFs, the major component of stroma in malignancies, play an essential role in tumor proliferation, progression, and treatment resistance [170,173]. Thus, CAFs are suggested to be a potential therapeutic target for the treatment of different tumor types including melanoma. Many drugs targeting CAFs have been developed and tested in preclinical and/or clinical studies. The most identified targets of CAFs are the fibroblast activation protein (FAP), vitamin D receptor (VDR), and platelet-derived growth factor receptor (PDGRF) [174,175,176].

FAP is a serine protease with dual enzymatic activities and is overexpressed in CAFs and in many other tumor types [177]. The G-protein-coupled receptor 77 (GPR77) is a potential FAP surface target and is specifically expressed in CAFs [174,175,176]. In addition to its important role in tumor development, the overexpression of FAP on CAFs is mostly associated with poor prognosis [178,179].

Accumulating evidence suggests that vitamin D does not only suppress cancer cells but also contributes to the modulation of tumor stromal cell genes and triggers tumor angiogenesis, progression, and metastasis [180,181]. These observations suggest that the vitamin D receptor is a promising target for the treatment of tumors such as melanoma. Figure 5 demonstrates the impact of CAF as a therapeutic target for melanoma treatment.

Several studies suggested that an important role exists for PDGF in the regulation of the recruitment and phenotypic character of the tumor stroma [114,182]. PDGF-BB has been shown to trigger the formation of growth-promoting stroma in melanoma [114,182]. Inhibition of vascular endothelial growth factor-A (VEGF-A) production promotes tumor cells to secrete PDGF-AA to attract stromal fibroblasts, which can be stimulated to produce VEGF-A and induce angiogenesis [183,184]. To that end, *PDGRF* is essential in promoting tumor growth by both direct growth stimulatory effects and promotion of angiogenesis and pericyte recruitment [185,186], and it is therefore a promising therapeutic target for tumor treatment.

### 4.2. Tumor-Associated Macrophages as Therapeutic Target

Tumor-associated macrophages (TAMs) have also emerged as therapeutic targets in melanoma treatment. TAMs belong to stromal cells and are abundant in the tumor microenvironment [187,188]. TAMs are mostly associated with poor clinical outcomes in cancer patients [189,190]. Accordingly, colony-stimulating factor 1 receptor (CSF1R) signaling has gained more attention as a therapeutic target. CSF1/CSF1R has been reported to play a central role in the proliferation, differentiation, and function of macrophages [191,192]. Therefore, inhibition of CSF1R signaling is expected to block the function of TAMs. Consequently, several inhibitors (PLX3397, JNJ-40346527, PLX7486, and ARRY-382) and neutralizing antibodies (RG7155, IMC-CS4, and FPA008) have been developed and tested for their clinical relevance as CSF1R inhibitor-based therapies [191,193]. Many preclinical and clinical investigations have demonstrated that inhibition of CSF1R results in the depletion of TAMs and microglia [194].

### 4.3. Tumor-Associated Neutrophils

Tumor-associated neutrophils (TANs) are also therapeutic targets in melanoma treatment. TANs originate from myeloid precursors and are the most abundant population of leukocytes as well as the first responders of innate immunity [174,195]. TANs are one of the most important stromal cells in the tumor microenvironment and play active roles in tumor progression and metastatic dissemination [196,197] TANs mediate their pro-tumor roles by stimulating ECM and inflammation in the tumor microenvironment [198]. TANs are characterized by their ability to release granules containing various proteases, such as matrix metalloprotease 9 (MMP-9) [199,200] and neutrophil elastase [201,202]. Consequently, TANs can remodel ECM and promote tumor invasion [190,203]. In addition to the production of proinflammatory cytokines/chemokines, TANs also produce immunosuppressive factors, including arginase 1 and TGF-β [204]. These immunosuppressive factors are mainly involved in the suppression of adaptive immunity [205] as well as in the release of HGF to promote tumor progression [206]. Thus, targeting TANs is expected to be a potential therapeutic strategy for tumor treatment [207]. One of the most promising targets is the chemokine receptor 2 (CXCR2), which is known to be a critical regulator for neutrophil mobilization [208]. Preventing the interaction between CXCR2 and its ligand (CXCL8) by small molecular inhibitors or antibodies has been shown to exert anti-tumor activities and improves the treatment efficacy of chemotherapy [208,209]. Several clinical trials such as SX-682 have been suggested as potent inhibitors of CXCR1/2. SX-682 has been tested for its clinical relevance in several studies [210]. SX-682 can block tumor cells by attracting myeloid-derived suppressor cells (MDSCs), which increases therapeutic efficacy when combined with immunotherapies [210].

## 5. Tumor-Promoting Chronic Inflammation as Therapeutic Target for Melanoma Treatment

Inflammation is a consequence of the innate immune response reacting to disturbed tissue homeostasis. Chronic inflammation is one of the common hallmarks of cancer and plays an essential role in the enhancement of tumor development and progression [211,212]. Thus, targeting inflammation is expected to be a promising approach for cancer therapy. Data obtained from a large population study revealed that aspirin is an anti-inflammatory drug found to significantly reduce cancer risk [213,214]. Both macrophages and tumor cells are characterized by their potency to produce proinflammatory cytokines and inflammatory mediators and thereby sustain tumor cell proliferation and survival [215,216], immune evasion [217], angiogenesis [218,219], ECM remodeling [220,221], metastasis [222], chemoresistance [218,223], as well as radio-resistance [224]. To that end, targeting the key mediators of proinflammatory pathways and/or the main regulators (e.g., NF-κB and STAT3) of inflammatory cytokines (e.g., IL-1, TNF, and IL-6) is expected to inhibit cancer-promoting inflammation. Unfortunately, few antibodies/inhibitors exhibited anti-tumor activities in preclinical studies; thus, only a few candidates are under investigation in early-stage clinical trials [96,225]. Therefore, the main challenge of targeting inflammation is how to develop selective anti-inflammatory approaches without impairing anti-tumor immunity.

Other components of the tumor microenvironment can function as targets for melanoma treatment. These include B lymphocytes, regulatory T cells (Treg), adipocytes, mesenchymal stem cells, and exosomes [226,227]. These tumor microenvironment components have been shown to influence tumor progression and therapeutic responses [228]. Tregs are characterized by the expression of the transcription factor fork head box protein p3 (Foxp3) that is involved in the suppression of anticancer immunity [229,230,231]. Consequently, the protective immunosurveillance of tumors can be impaired, resulting in the loss of effective antitumor immune responses. Functionally, the tumor microenvironment can trigger the suppression of Tregs by the upregulation of immune checkpoint proteins. Thus, targeting immune checkpoint molecules (e.g., CTLA-4, TIGIT, PD-1, and GITR) on Tregs may have a therapeutic impact on the treatment of melanoma.

The role of inhibitory receptors in the regulation of both innate and adaptive immunity in chronic viral infections and cancer has been studied [232,233]. Chronic antigen stimulation mainly results in the modulation of T cell dysfunction and the upregulation of inhibitory receptors such as programmed cell death-1 (PD-1) and T cell immunoreceptor with immunoglobulin and immunoreceptor tyrosine-based inhibitory motif (ITIM) domain (TIGIT) [234,235]. In addition to the expression of the ligands of the inhibitory receptors by tumor cells, the tumor microenvironment contains the required antigen-presenting cells (APCs) [236,237]. TIGIT has been reported to play a critical role in the reduction of both adaptive and innate immunity against tumors [234,235]. The clinical relevance of monoclonal antibodies targeting the inhibitory receptors has been reported in several studies [238].

T lymphocyte-associated antigen 4 (CTLA-4) is one of the first identified inhibitors of immune checkpoint on Tregs. Targeting CTLA-4 by anti-CTLA-4 antibodies has been shown to block the tumor suppressive function of Tregs and ultimately to release the cytotoxicity function of effector cells.

Programmed cell death 1 (PD-1) signaling is known to be hijacked by cancer cells to escape immune surveillance [239,240,241]. The intrinsic expression of PD-1 has been reported to contribute to the development of tumor monoresistance [242,243]. In melanoma cells, the activation of PD-1 by its ligand PD-L1 has been shown to trigger the activation of downstream mammalian targets of rapamycin signaling leading to tumor growth [244]. Thus, targeting the PD-1/PD-L1 axis has shown enormous success in a variety of human cancers [245,246]. Due to its durable tumor regression and prolonged stabilization of disease in patients with advanced cancers, antibody-mediated blockade of PD-L1 is clinically relevant.

In the last two decades, the treatment of a variety of malignancies based on immune checkpoint modulation has been promising compared to available therapeutic modalities. However, checkpoint modulation has been reported to be less therapeutically effective in cancers with an immunosuppressive microenvironment [247,248]. Although the advent of immunomodulatory agents has led to improved responses in tumor patients, as evidenced by achieving long-lasting tumor remission [249,250], many exhibit brief or no response to available immunomodulatory agents [251,252]. Thus, the development of alternate therapeutic strategies is of great interest. In recent years, the modulation of the tumor microenvironment, in the context of the local metabolites, has been suggested as a promising strategy in cancer immunotherapy [253]. For example, live tumor-targeting bacteria have emerged as a treatment for solid tumors, compared with immunotherapy and targeted therapy [254]. Likewise, the clinical investigation of live engineered bacteria for metabolic modulation has been reported [255].

Oncolytic viruses have also been suggested as a promising alternative therapy for cancer treatment, particularly for refractory cancers with a 5-year survival rate of 5%, such glioblastoma [256]. While viral-mediated oncolysis has been hypothesized to spread to all cancer cells within the tumor, this has not been shown in clinical studies so far. Clinical data revealed that the development of an antiviral immune response and limited antitumor immunity limit the efficiency of virotherapy when utilized as a monotherapy [257,258]. Apart from the abovementioned limitations of virotherapy, the mechanisms of viral infection, replication, and tumor necrosis have the potential to destruct the immunosuppressive tumor microenvironment and ultimately enhance T cell reactivity against cancer neo-antigens [259].

The advantage of oncolytic virotherapy over checkpoint-protein-based immune therapy is attributed to the ability of oncolytic virotherapy to circumvent the immune evasion mechanisms of the tumor [260,261]. Oncolytic virotherapy can also improve the treatment outcome of tumor patients by the stimulation of host immune system and/or direct lysis of tumor cells.

## 6. Conclusions

The impact of the tumor microenvironment in melanoma development and therapy has gained more attention in academic research and the pharmaceutical industry. The functional characterization of tumor microenvironment components elucidates the tumor microenvironment as a promising therapeutic target for melanoma treatment. Of note, targeting the individual components of the tumor microenvironment alone may be insufficient for executing broad and sustainable therapeutic efficacy in melanoma patients. Likewise, many clinical trials targeting the tumor microenvironment have not shown promising efficacy in melanoma patients. Immune-checkpoint-blockade-based therapeutics, however, have shown promise for treatment success in patients with melanoma metastasis. The reported success of immunotherapy in cancer patients is attributed to deciphering the fundamental mechanisms of T cell activation and inhibition. Thus, understanding the fundamental components underlying the tumor microenvironment, such as fibroblasts and macrophages, may facilitate the discovery and development of novel drugs targeting the tumor microenvironment. Importantly, the high heterogeneity of the tumor microenvironment has allowed the research and drug industry to develop reliable biomarkers to guide tumor-microenvironment-targeted therapies. As a whole, the establishment of combinatory therapy with the aim of maximizing treatment efficacy benefits more patients.

## Figures and Tables

**Figure 1 cancers-15-03147-f001:**
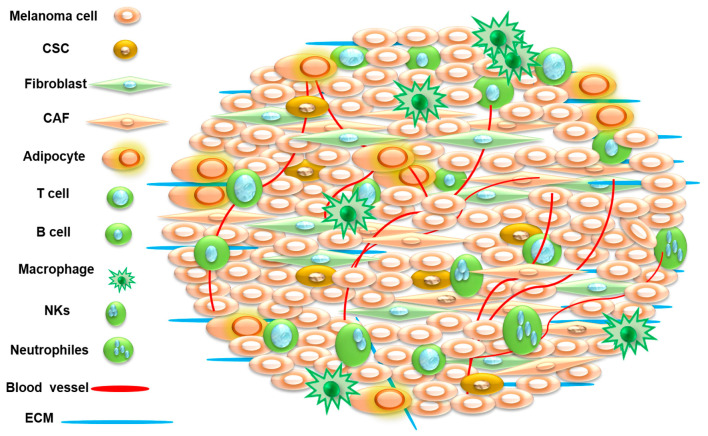
Schematic diagram of the microenvironment in solid tumors. The tumor microenvironment is a complicated ecosystem of heterogeneous components including tumor cells, stromal cells, and variable types of immune cells, soluble molecules, and extracellular matrix (ECM) components. Stromal cells include endothelial cells, fibroblasts/cancer-associated fibroblasts (CAFs), cancer stem cells, as well as immune cells such as macrophages/tumor-associated macrophages (TAMs), neutrophils/tumor-associated neutrophils (TANs), natural killer cells (NKs), and T cells and B cells. Non-cellular components including ECM exist in a three-dimensional scaffold of extracellular macromolecules, proteins, and polysaccharides that provides structural and biochemical support to cells and soluble molecules (chemokines, cytokines, growth factors, and extracellular vesicles).

**Figure 2 cancers-15-03147-f002:**
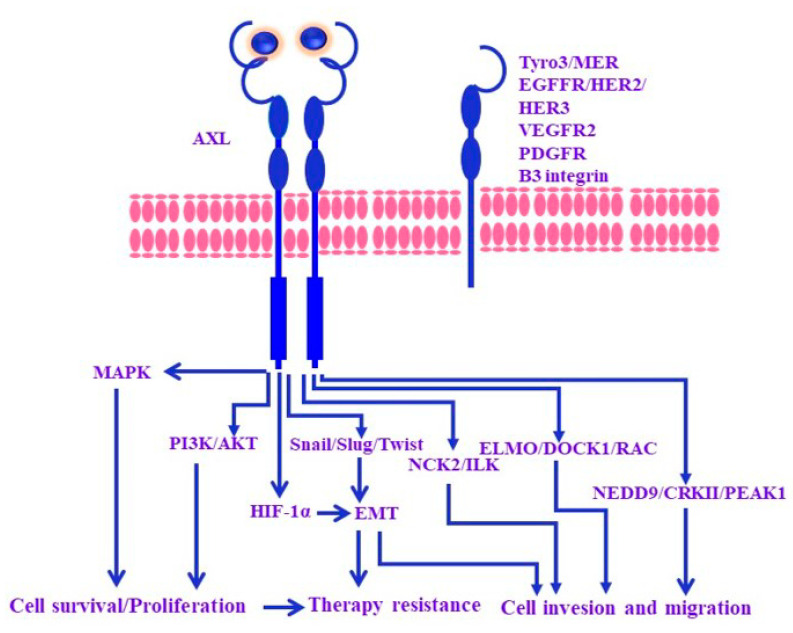
Outline of AXL-mediated downstream signaling pathways. The majority of reported AXL activation by GAS6 results from the homodimerization of the AXL receptor or heterodimerization of AXL with TYRO3. AXL-stimulated MAPK and PI3K/AKT signaling results in cell survival and proliferation. Likewise, AXL-stimulated HIF-1α-EMT triggers therapy resistance. AXL-stimulated Snail/Slug/Twist-EMT, NCK2/ILK, ELMO/DOCK1/RAC, und NEDD9/CRKII/PEAK1 trigger tumor invasion and migration.

**Figure 3 cancers-15-03147-f003:**
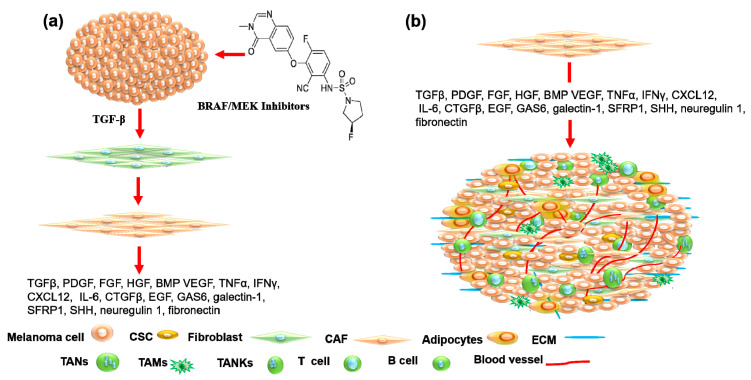
Schematic diagram of tumor-induced activation of fibroblasts and their biological consequences. (**a**) Conversion of normal fibroblasts into cancer-associated fibroblasts (CAFs) by melanoma-cell-released transforming growth factor-β (TGF-β). (**b**) CAF-released factors include TGF-β, platelet-derived growth factor (PDGF), fibroblast growth factor (FGF), HGF, bone morphogenetic proteins (BMP), vascular endothelial growth factor (VEGF), tumor necrosis factor α (TNFα), interferon γ (IFNγ), CXC-motif-chemokine 12 (CXCL12), interleukin 6 (IL-6), connective tissue growth factor β (CTGFβ), endothelial growth factor (EGF), growth arrest-specific 6 protein (GAS6), galectin-1, secreted frizzled-related protein-1 (SFRP1), sonic hedgehog protein (SHH), neuregulin 1, and fibronectin. These CAF-secreted factors are involved in the modulation of melanoma progression, metastasis, and treatment resistance.

**Figure 4 cancers-15-03147-f004:**
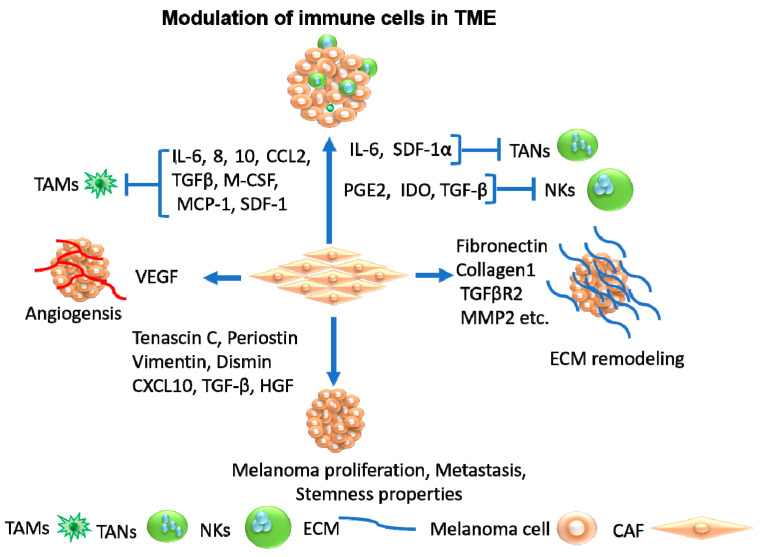
The mechanisms by which CAF-released factors trigger tumor angiogenesis, ECM remodeling, suppression of adaptive immunity, melanoma progression, metastasis, treatment resistance, and stemness properties. The induction of tumor angiogenesis is mediated by CAF-released VEGF. The suppression of adaptive immunity in the tumor immune microenvironment is attributed to the inhibition of tumor-associated macrophages (TAMs) by CAF-released IL-6, 8, 10, CCL2, TGF-β, macrophage colony stimulated factor (M-CSF), monocyte chemoattractant protein-1 (MCP-1), stromal cell-derived factor 1 (SDF-1), as well as the inhibition of tumor-associated neutrophiles (TANs) by CAF-released IL-6 and SDF-1α and the inhibition of natural killer cells (NKs) by CAF-released prostaglandin E2 (PGE2), indolamin-2,3-dioxygenase (IDO), and TGF-β. The remodeling of the ECM is mediated by CAF-released fibronectin, collagen1, TGFβR2, and matrix metalloproteinase-2 (MMP2). The induction of melanoma proliferation, metastasis, and stemness properties is mediated by CAF-released tenascin C, periostin, vimentin, dismin, chemokine (C-X-C motif) ligand 10 (CXCL10), TGF-β, and HGF.

**Figure 5 cancers-15-03147-f005:**
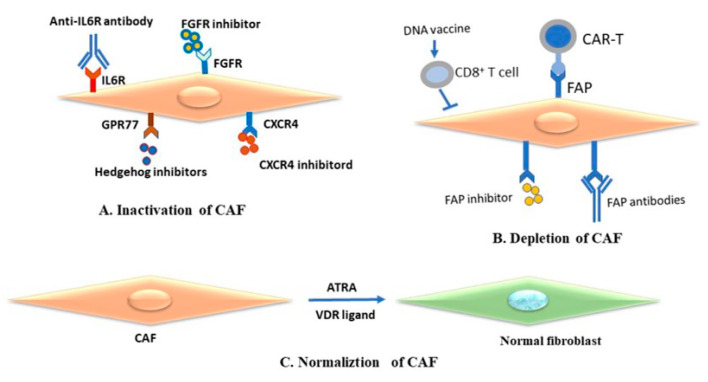
Proposed strategies for melanoma treatment by targeting cancer-associated fibroblasts (CAFs) in tumor microenvironment. (**A**) Inactivation of tumor-promoting function of CAFs by targeting crucial signaling pathways by the specific inhibitors of hedgehog, FGFR, and CXC4R. (**B**) Depletion of CAFs by targeting CAF-specific markers, such as FAP. (**C**) Normalization of tumor-suppressive state with small molecules such as ATRA or VDR ligands. FAP: fibroblast activation protein; CAR: chimeric antigen receptor; ATRA: all-trans retinoic acid; VDR: vitamin D receptor; FGFR: fibroblast growth factor receptor.

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
