# Peer review of "Tumor Microenvironment as a Therapeutic Target in Melanoma Treatment"

_cancers, 2023, doi:10.3390/cancers15123147_

Round 1
Reviewer 1 Report
In this review, the authors described the common mechanisms of cutaneous melanoma resistance, emphasizing aspects of the tumor microenvironment, and further pointed to potential opportunities for intervention based on the interactions of resistant tumor cells and interacting microenvironmental cells. The text is collegial and well-balanced but does not bring novel insights to the area, saturated with similar reviews. The approach lacks a mechanistic understanding at the molecular level. Although molecular effectors and cellular processes were identified throughout the text, signal transduction pathways and potential molecular circuitries, which could be targeted, were not discussed in detail. The authors are encouraged to revise this issue to benefit their work's completeness.
Primary suggestions for improvement include the following.
1. Expand on the mechanism of AXL and other MER-TKs. What are the molecular bases for their activation in cutaneous melanomas? Which cells express these molecules in melanoma tumors?
2. Critical literature about MITF is missing.
3. Please discuss the impact of senescence in melanoma and microenvironmental cells. Is there a role for senolysis in therapy?
4. Please discuss autophagy in treatment resistance and the release of extracellular vesicles from melanoma cells. What is the impact of the released vesicles on the interactions of the distinct intratumoral cell subpopulations?
5. There are novel strategies to interfere with the immune synapse besides the discussed immune checkpoint blockers. Please extend the presentation of these strategies.
Minor points.
1. The authors focused on cutaneous melanomas. This fact needs to be stated clearly in the introduction of their work.
2. Please revise the wording carefully.
Please revise the wording carefully. So misspelled words were found in the text.
Author Response
Editor
Cancers
Dear Editor,
Thank you very much for the encouraging comment regarding our Manuscript: “Tumor Microenvironment as a Therapeutic Target in Melanoma Treatment”. As required, please find enclosed our response “Point-for-Point” to the valuable comments of Reviewers.
On behalf all coauthors
Authors ‘response to Reviewer 1

Reviewer 2 Report
The selected content is novel and the figures are beautiful and uniform. The logic is also good. I’d recommend its publication after revisions.
Major:
1. Is there any unique difference between tumor microenvironment (TME) of melanoma and other tumors?
2. Lacking description of other TME factors such as low pH, full H2O2 and GSH, and lack of oxygen (please refer to: 1. 10.1002/advs.202103836; 2. 10.1002/ange.201903981).
3. The “4. Tumor microenvironment as therapeutic target in melanoma treatment” section has too little text since it should be the most important part in the paper according to the title, and I’d suggest using sub-heading for each target of TME.
4. Please draw at least two more diagrams in part 4 and 5, as these are the most important parts but without diagrams.
5. In conclusion part, the authors always talk about tumor and TME, but not about the specific melanoma, that’s unreasonable.
Minor:
6. There are too few keywords, maybe you can add more, like Targeted therapy or Chronic inflammation.
7. In Fig1, the macrophage should be drawn bigger.
8. Please list the full name of BRAF and MEK.
9. MIFT should be MITF in line 133.
Minor editing of English language required about the grammar
Author Response
Editor
Cancers
Dear Editor,
Thank you very much for the encouraging comment regarding our Manuscript: “Tumor Microenvironment as a Therapeutic Target in Melanoma Treatment”. As required, please find enclosed our response “Point-for-Point” to the valuable comments of Reviewer 2
On behalf all couathors.
Authors ‘response to Reviewer 1

Round 2
Reviewer 1 Report
The revised version is much improved and conceptually accurate. Although I still miss perceiving the scientific contribution of the authors in the text, this text will be useful to the journal's readership.